# Treatment of Dyslipidemia in Patients with Type 1 Diabetes Mellitus: A Review of Current Evidence and Knowledge Gaps

**DOI:** 10.3390/ijms26178558

**Published:** 2025-09-03

**Authors:** Viviana Elian, Alina Dorita, Daniela Stegaru, Dragos Vinereanu

**Affiliations:** 1Diabetes, Nutrition and Metabolic Disease Unit, University of Medicine and Pharmacy Carol Davila, 5-7 Ion Movila Street, 0202475 Bucharest, Romania; stegaru.daniela@gmail.com; 2Diabetes, Nutrition and Metabolic Disease Unit, National Institute of Diabetes, Nutrition and Metabolic Disease Prof. N. C. Paulescu, 020475 Bucharest, Romania; doritaalina18@gmail.com; 3Department of Cardiology and Cardiovascular Surgery, University of Medicine and Pharmacy Carol Davila, 169 Splaiul Independenței, 050098 Bucharest, Romania; vinereanu@gmail.com; 4Department of Cardiology, University and Emergency Hospital, 050098 Bucharest, Romania; 5SEARCH-Vasc Centre of Excellence, 050098 Bucharest, Romania

**Keywords:** type 1 diabetes, dyslipidemia, lipid-lowering therapy, statins, ezetimibe, PCSK9 inhibitors, cardiovascular disease, clinical guidelines

## Abstract

Type 1 diabetes (T1D) is a chronic condition with an increasing prevalence worldwide and a significant improvement in life expectancy in the last decades. T1D confers an increased risk of cardiovascular events, driven by elevated LDL cholesterol (LDL-C) and qualitative lipoprotein abnormalities, such as dysfunctional HDL and smaller, denser LDL-C. Lipid-lowering outcome trials have overwhelmingly focused on type 2 diabetes or the general population, resulting in very limited T1D-specific evidence. Recommendations from major medical associations (ADA, ESC/EAS, ACC/AHA, ISPAD) create additional ambiguity regarding the treatment of dyslipidemia in T1D. This review synthesizes the available evidence on dyslipidemia management in T1D, including published observational cohorts, randomized controlled trials, and international guideline recommendations from January 2000 to June 2025. LDL-C remains the primary modifiable risk factor. Each 1 mmol/L increase is associated with 35–50% greater cardiovascular (CV) risk in T1D cohorts. Statin therapy reduces CV risk by up to 25% in patients with diabetes; however, evidence remains limited in patients with T1D. Ezetimibe provides an additional 18% LDL-C lowering and a 14% event reduction in mixed-diabetes trials, while PCSK9 inhibitors offer a potent 40–60% LDL-C reduction and an 18% MACE reduction. The uptake of statins in eligible adults with T1D remains below 50%. Statins remain the cornerstone of dyslipidemia management in T1D, with emerging evidence supporting ezetimibe and PCSK9 inhibitors. The heterogeneity across international guidelines and the scarcity of T1D-specific outcome data underscore the need for targeted research and evidence-based strategies.

## 1. Introduction

The incidence of type 1 diabetes (T1D) is globally rising by 2 to 5% yearly, with an estimated prevalence of almost 9.5 million persons in 2025 [1]. Advancements in the treatment of T1D (new rapid and ultra-rapid insulin analogs, slow insulin with longer duration of action, insulin pumps, blood glucose sensors) have significantly improved the quality of life and overall life expectancy, primarily by limiting the progression of microvascular complications and by preventing acute metabolic complications. However, as life expectancy increased, cardiovascular (CV) complications emerged, highlighting the need for targeted preventive strategies and specialized therapeutic interventions.

Patients with T1D are at an increased risk of CV events and mortality, even in young adults, most likely due to multiple cardiovascular risk factors such as dyslipidemia, autonomic neuropathy, nephropathy, hypertension, high glycemic values, insulin resistance, obesity, smoking, and low physical activity [2]. Subclinical atherosclerosis, such as increased arterial stiffness and increased carotid and aortic intima–media thickness (IMT), is highly prevalent and associated with poor glycemic control, insulin resistance, body mass index, elevated blood pressure, and lipid levels in adolescents and children with T1D, as found in the SEARCH study [2,3]. However, cardiovascular disease (CVD) risk remains increased even in well-controlled T1D patients, without additional CV factors, suggesting that other factors may be involved [3].

Pathophysiology of lipid disorders in T1D fundamentally differs from T2D and insulin-resistance conditions. Thus, in T1D, absolute insulin deficiency impairs lipoprotein lipase activity, leading to decreased clearance of triglyceride-rich lipoproteins (chylomicrons and very-low-density lipoproteins (VLDLs)), rather than increased hepatic VLDL production, which is characteristic of insulin resistance [4]. In type 2 diabetes (T2D), insulin resistance stimulates hepatic lipogenesis through persistent Sterol Regulatory Element-Binding Protein 1c (SREBP-1c) activation and increased free fatty acid from adipose tissue lipolysis, resulting in the classic triad of hypertriglyceridemia, low high-density lipoprotein cholesterol (HDL-C), and small dense LDL-C particles [5,6,7].

CV risk emerges earlier in T1D through multiple insulin deficiency-independent pathways, including hyperglycemia-induced advanced glycation end product (AGE) formation, enhanced myelopoiesis with calprotectin release stimulating AGE receptors (RAGE), with modified glycated and oxidized LDL particles, and accelerated atherosclerosis [8]. Chronic hyperglycemia induces AGE formation that leads to NADPH oxidase activation, mitochondrial reactive oxygen species generation, and subsequent endothelial dysfunction through decreased nitric oxide bioavailability [9]. This process is amplified by protein kinase C (PKC) activation and stimulated polyol pathways, generating oxidative stress and accelerating atherogenesis [10]. The autoimmune process involved in T1D contributes to an additional inflammatory burden, with calprotectin release from activated neutrophils promoting myelopoiesis through bone marrow RAGE [11,12].

Peripheral hyperinsulinemia from subcutaneous insulin administration can lead to qualitative HDL-C abnormalities and altered lipoprotein composition that may compromise some cardioprotective functions [2]. These molecular changes explain why even well-controlled patients with T1D exhibit qualitative lipoprotein abnormalities that persist independently of glycemic status [13,14].

Despite the significant CV burden imposed by dyslipidemia in T1D, current guidelines largely extrapolate recommendations from data derived from T2D. This approach is primarily due to the pronounced scarcity of dedicated randomized controlled trials (RCTs) investigating lipid-lowering therapies and their long-term outcomes in T1D. Consequently, a critical knowledge gap remains regarding the optimal, evidence-based strategies for treating dyslipidemia in T1D, tailored to the unique pathophysiological and clinical characteristics of the disease. This review aims to address this gap by synthesizing the existing evidence, thereby clarifying what is currently known, while highlighting areas where further T1D-specific research is urgently needed.

## 2. Results

Published data revealed a substantial increase in cardiovascular risk with each 1 mmol/L (38.7 mg/dL) rise in LDL-C, as highlighted by data from the Swedish National Diabetes Registry [15]. Landmark trials, like the Diabetes Control and Complications Trial/Epidemiology of Diabetes Interventions and Complications (DCCT/EDIC) [16] and the Pittsburgh Epidemiology of Diabetes Complications (EDC) study [17], consistently identified LDL-C as a significant risk factor for CVD and Major Atherosclerotic Cardiovascular Events (MACEs) in T1D. The Pittsburgh EDC study specifically suggests maintaining an LDL-C level less than 100 mg/dl as optimal for primary CVD prevention in T1D [18].

Both lifestyle modifications and targeted pharmacotherapy represent current interventions for dyslipidemia in T1D. It should be noted that most studies and guidelines from which these recommendations are elaborated are conducted on patients with T2D or patients with diabetes without differentiation between the two types and then extrapolated to patients with T1D.

**Lifestyle interventions** are recommended as first-line therapy for dyslipidemia in patients with diabetes. According to the American Diabetes Association (ADA) and American Heart Association (AHA) guidelines, medical nutrition therapy typically reduces LDL cholesterol by 15–25 mg/dL in patients with diabetes [19,20].

A randomized clinical trial that enrolled 58 patients with T1D examined the effects of a low-fat vegan diet versus a portion-controlled diet and found that the vegan group showed a significant reduction in total cholesterol (−32.3 mg/dL) and LDL-C (−18.6 mg/dL). In contrast, the portion-controlled group showed small (−10.9 mg/dL) or no significant changes for total cholesterol and LDL-C, respectively [21].

A study from China, involving 99 individuals with type 1 diabetes, identified specific dietary patterns that significantly influence LDL-C levels. Participants in the highest tertile of an unfavorable dietary pattern had significantly higher LDL-C compared to those in the lowest tertile, with a mean difference of 14 mg/dL [22].

The FinnDiane Study indicates a U-shaped relationship between dietary sodium intake and mortality in 2807 adults with T1D followed up for 10 years, where low sodium intake (<2.3 g/day) and high sodium intake (>5.0 g/day) are both linked to increased all-cause mortality (hazard ratios of 1.35 and 1.44, respectively) [23].

DASH (Dietary Approaches to Stop Hypertension) or Mediterranean diet are the only nutritional interventions recommended by guidelines as a first-line approach for decreasing LDL-C [21]. These dietary patterns can positively impact lipid profiles but have not been demonstrated through a randomized trial to have a direct impact on reducing cardiovascular risk in patients with type 1 diabetes.

**Statins** are the first-line therapy for LDL-C in patients with T2D, as demonstrated by the CARDS (Collaborative Atorvastatin Diabetes Study) [24] and HPS (Heart Protection Study) [25] trials. Statin therapy has been shown to reduce CV risk by 20% in the general population [26], for every 1 mmol/L (39 mg/dL) decrease in LDL-C, and by 25% in patients with diabetes (including both T1D and T2D), even in those with a baseline LDL-C less than 116 mg/dL [27].

A prospective cohort study from Swedish registries included 24,230 persons with T1D without a history of CVD with a mean follow-up of 6 years, and 18,843 were untreated and 5387 were treated with lipid-lowering therapy (LLT) (97% statins) [28]. The LLT group had a 22–44% decrease in all-cause mortality in primary prevention, with a reduction in the incidence of CV death (40%), all-cause death (44%), CVD (23%), stroke (44%), CHD, and acute myocardial infarction (22%) [28].

The AdDIT trial enrolled 443 high-risk adolescents with T1D and 400 low-risk adolescents with T1D, randomized into four groups: Quinapril, Atorvastatin, both, or placebo. Every group comprised approximately 200 patients (100 high-risk and 100 low-risk). It found that 3–4 years of statin therapy did not improve carotid intima–media thickness (hazard ratio (HR) 0.57; 95% confidence interval (CI), 0.35–0.94) or other cardiovascular markers. Atorvastatin significantly decreased LDL-C levels by 21.3%, total cholesterol by 23.7%, and non-HDL-C by 23.5% [29]. The AdDIT trial, despite being the largest pediatric T1D cardiovascular intervention study, demonstrated only modest effects on carotid intima–media thickness without clear clinical benefit [29].

In a meta-analysis performed by de Vries et al., which included nine randomized controlled trials, 9151 patients received either a standard statin dose/placebo or an intensive statin dose/placebo. The results showed that the standard use of statins in the secondary prevention of major cardiovascular or cerebrovascular events in patients with diabetes is associated with a 15% significant relative risk reduction [30]. Intensive treatment provided an additional 9% (RR 0.90, 95% CI 0.85–0.96) reduction in risk. Secondary endpoints of the analysis of a standard dose of statin compared to placebo achieved a significant relative risk reduction of 33% for fatal and non-fatal stroke and a non-significant relative risk reduction of 27% for fatal and non-fatal MI and of 22% for all-cause mortality [30].

A South Korean study that enrolled 11,009 patients with T1D over the age of 20 years, without CVD before T1D diagnosis, for a mean follow-up of 10 years, found that any statin use (atorvastatin, fluvastatin, lovastatin, pitavastatin, pravastatin, rosuvastatin, or simvastatin) reduced cardiovascular risk by 24%. Statin use was significantly associated with a reduction of 26% in both ischemic stroke and myocardial infarction [31]. Moreover, MACEs decreased by 27% in the first 1 to 3 years of statin treatment and by 40% for an exposure of more than 3 years [31]. A worrisome problem raised by this study was that less than half of the patients were on statins after 10 years. Therefore, better adherence is needed, especially in high-risk T1D patients.

The PADIT study, conducted in 51 children and adolescents (10–21 years old) with type 1 diabetes, investigated the effects of adding 20 mg of atorvastatin on arterial stiffness and endothelial function. Atorvastatin significantly reduced LDL-C by 25%. Radial artery tonometry and reactive hyperemia were performed at 8 and 12 weeks of follow-up. The primary analysis on primary objectives failed to demonstrate an associated improvement in vascular parameters with a potential benefit; however, the secondary analysis, focusing solely on reducing arterial stiffness, showed a possible benefit (*p* = 0.06). The small sample size might have altered these results [32].

The DIATOR trial, conducted on 89 patients (aged 18–39) with recent-onset T1D and at least one islet autoantibody, investigated the effect of atorvastatin (80 mg/day) on the decline of pancreatic beta-cell function compared to placebo. There was no statistically significant difference in main fasting and stimulated serum C-peptide levels (0.30 vs. 0.20 nmol/L, *p* = 0.40, and 0.71 vs. 0.48 nmol/L, *p* = 0.36, respectively) between the atorvastatin and placebo groups at 18 months. Still, the atorvastatin group showed a slower decline in both fasting and stimulated C-peptide levels compared to placebo (19% vs. 46%). Moreover, in the atorvastatin group, median baseline levels of total cholesterol, LDL-C, and triglycerides significantly decreased by 32.2%, 52.3%, and 26.0%, respectively (*p* < 0.001 for all), while HDL-C increased by 16.2% (*p* < 0.001) [33].

The PLANET I study, conducted on 353 adults with T1D (47 patients) or T2D and proteinuria (UPCR 500–5000 mg/g), all on stable ACE-inhibitors or ARBs, followed the effect of atorvastatin 80 mg, rosuvastatin 10 mg, and rosuvastatin 40 mg on UPCR. The results showed that atorvastatin 80 mg significantly reduced UPCR (ratio, 0.87; *p* = 0.033), whereas rosuvastatin did not substantially mitigate UPCR, regardless of the dose. Post hoc analysis (PLANET I and II) showed that atorvastatin reduced UPCR appreciably more than both rosuvastatin doses (vs. 10 mg: −15.6%, *p* = 0.043; vs. 40 mg: −18.2%, *p* = 0.013) [34].

A meta-analysis conducted on nineteen randomized, double-blind trials comparing statins vs. placebo (123,940 participants) and four trials comparing more vs. less intensive statin therapy (30,724 participants) has examined the risk of developing new-onset diabetes, glycemic control in people with existing diabetes, and whether these effects vary by statin intensity, patient characteristics, or over time. The risk of new-onset diabetes was increased by statin therapy by 10% with low-/moderate-intensity statins and by 36% with high-intensity statins, corresponding to absolute annual excesses of 0.12% and 1.27%, respectively. It is essential to mention that most cases (approximately 62%) occur in people already near the diagnostic threshold for diabetes. Despite these risks, the cardiovascular benefits of statins outweigh the glycemic drawbacks; statin therapy reduces cardiovascular risk by about 25% for every 1 mmol/L reduction in LDL-C, regardless of the presence of diabetes [35].

This data suggests that information regarding the benefit of statins in patients with T1D is promising but limited, underlining the importance of conducting more studies in this population (Table 1). 

**Ezetimibe** selectively inhibits the intestinal absorption of cholesterol, resulting in lower cholesterol levels in the portal blood and liver, which stimulates the upregulation of LDL-C receptors on hepatocytes, ultimately leading to a reduction in serum LDL-C levels [3].

The IMPROVE-IT trial, conducted on 18,144 patients recently diagnosed with acute coronary syndrome (ACS) and LDL-C between 50 and 125 mg/dL, demonstrated superiority of ezetimibe/statin combination therapy over statin monotherapy, both in the general population and in a sub-analysis of patients with diabetes (4933 patients without specifying the T1D population) [36]. Absolute risk reduction in the composite endpoint (MACE, CV death, and stroke) was greater in the diabetic subgroup vs. individuals without diabetes (5.5%, HR: 0.85; 95% CI: 0.78–0.94 vs. 0.7% HR: 0.98; 95% CI: 0.91–1.04, *p* = 0.02) [36].

In the diabetes subgroup of the RACING [37] study, the efficacy and safety of 10 mg rosuvastatin as moderate-intensity statin plus 10 mg ezetimibe were compared with 20 mg rosuvastatin as high-intensity statin monotherapy in a total of 1398 patients with diabetes and ASCVD, representing 37% of the trial population. Combined therapy was significantly (*p* < 0.001) superior to monotherapy in reducing LDL-C at 1, 2, and 3 years: 81.0%, 83.1%, and 79.9% of patients in the ezetimibe combination therapy group, versus 64.1%, 70.2%, and 66.8% of patients in the statin monotherapy group. Incidence of primary composite endpoint (CV death, myocardial infarction, coronary revascularization, HHF, or non-fatal stroke) did not differ significantly between the two groups (10.0% vs. 11.3%, HR: 0.89; 95% CI: 0.64–1.22; *p* = 0.460) [37]. This is consistent with the results for the overall RACING trial population, which included 3780 patients with established ASCVD, where the incidence of the primary composite endpoint occurred in 9.1% of patients receiving combination therapy and 9.9% of those on high-intensity statin (HR = 0.92, 90% CI: 0.75–1.13, *p* < 0.0001) [38].

The most potent effect of ezetimibe in patients with diabetes may be attributed to its influence on postprandial hyperlipemia, a characteristic of the lipid phenotype in diabetes [39]. An exploratory, non-randomized, crossover study revealed that ezetimibe was more effective in lowering LDL-C levels in patients with T1D compared to those with T2D; meanwhile, within the T1D group, ezetimibe lowered LDL-C more than statins [40]. The greater efficacy observed in T1D may be attributable to evidence indicating enhanced cholesterol absorption and reduced cholesterol synthesis in these patients [41]. More data on available studies using ezetimibe are available in Table 2.

**PCSK9 inhibitors**. This newer class of lipid-lowering drugs acts by inhibiting PCSK9 (Proprotein Convertase Subtilisin/Kexin Type 9)-mediated degradation of LDL-C receptors in hepatocytes, thereby increasing LDL receptor expression. In young patients with T1D, PCSK9 concentrations are elevated and positively associated with triglycerides, total cholesterol, LDL-C, and HbA1c. Poor or suboptimal glycemic control is linked to higher PCSK9 levels and a greater prevalence of small, dense LDL-C (sdLDL-C) particles, whereas HbA1c values <7.5% are associated with lower circulating levels of PCSK9 and sdLDL-C [42,43]. Even if PCSK9 inhibitors demonstrate strong lipid-lowering efficacy, achieving an average LDL-C reduction of 47.8% versus placebo in individuals with T1D [44], the evidence for CVD prevention in T1D is limited, as 97% of participants in key studies had T2D. Therefore, PCSK9 inhibitors are primarily used for secondary prevention in patients with T2D, with a history of CVD, who are unable to achieve optimal LDL-C levels on maximally tolerated statin therapy [45].

The FOURIER trial was conducted on 27.564 patients, of which 39.3% had T2D and 0.7% had T1D, examining a composite of CV death, myocardial infarction, stroke, hospitalization for unstable angina, or coronary revascularization of patients treated with evolocumab versus placebo added to statin therapy [46]. Individuals with T1D had the highest CV event rate in the placebo group at 20.4% compared to 15.2% in those with T2D and 11.0% in participants without diabetes. Treatment with evolocumab reduced the risk of major cardiovascular events across all groups, with hazard ratios of 0.87 (95% Cl 0.79–0.96) and absolute risk reduction of 1,3% for non-diabetes, 0.84 (95% Cl 0.75–0.93) and 2.5% for those with T2D, and 0.66 (95% Cl 0.32–1.38) and 7.3% for those with T1D. Notably, the absolute risk reduction was most significant in the T1D group, despite the small sample size and wide confidence interval [46,47]. This underscores the fact that individuals with T1D and ASCVD have a substantially higher cardiovascular risk. Still, evolocumab may offer significant benefit; however, due to the small number of patients with T1D included in the study, further randomized trials explicitly focused on T1D populations are needed to confirm these findings. The FOURIER trial’s T1D cohort included only 197 patients, resulting in wide confidence intervals (HR 0.66, 95% CI 0.32–1.38) that, while suggesting potential benefit, lack definitive statistical significance [46,47].

The ODYSSEY OUTCOMES study, conducted in 18,924 individuals (28.8% with diabetes, without specifying the percentage of patients with T1D, 43.6% prediabetes, and 27.7% normoglycemic), evaluated the effect of alirocumab on reducing cardiovascular risk in adults with a recent acute coronary event who were undergoing statin therapy. It reduced LDL-C levels to approximately 31 mg/dL within four months across all patient groups. This lipid-lowering effect translated into a meaningful reduction in MACEs, with a 15% relative risk reduction with an HR of 0.85 (95% CI: 0.78–0.93) overall. Patients with diabetes experienced the most significant relative risk reduction; alirocumab lowered MACE risk by 16% (HR = 0.84; *p* = 0.002), while those with prediabetes experienced a 14% reduction (HR = 0.86; *p* = 0.009) [48]. In contrast, the reduction in euglycemic individuals was minor and not statistically significant, with a 7% decrease in MACE risk (HR = 0.93; *p* = 0.25). These findings suggest that individuals with impaired glucose metabolism may derive greater cardiovascular benefit from PCSK9 inhibition following acute coronary syndrome. Additionally, alirocumab lowered lipoprotein(a) levels by a median of 5.0 mg/dL, and each 1 mg/dL reduction in lipoprotein(a) was associated with a 0.6% reduction in MACE risk (HR 0.994; *p* = 0.0081), independent of LDL-C changes [48].

A post hoc analysis of the ODYSSEY DM-DYSLIPIDEMIA trial evaluated the lipid-lowering efficacy of alirocumab in 186 individuals with T2D, elevated triglycerides (≥200 mg/dL), and low HDL cholesterol (<40 mg/dL for men, <50 mg/dL for women), all receiving maximally tolerated statin therapy. Participants were randomized into two groups, alirocumab or conventional care (ezetimibe, fenofibrate, or no additional LLT), for 24 weeks. Alirocumab significantly reduced non-HDL cholesterol by 35%, ApoB by 34.7%, LDL-C by 47.3%, LDL-C particle number by 40.8%, and lipoprotein (a) by 29.9% compared to usual care (all *p* < 0.0001), while it increased HDL-C by 7.9% (*p* < 0.05); however, triglyceride reductions were modest and not statistically different [49].

ODYSSEY DM–INSULIN study was conducted on individuals with T1D or T2D (76 and 441 patients, respectively) who are at high CV risk and have hypercholesterolemia not adequately controlled by the maximum tolerated statin therapy [50]. Among patients with T2D, LDL-C decreased by 49.0%, while among patients with T1D, it decreased similarly, by 47.8%, both compared to placebo (*p* < 0.0001). Additionally, alirocumab led to a significant reduction in non-HDL cholesterol, apoB, and lipoprotein (a). By week 24, 76.4% of patients with type 2 diabetes (T2D) and 70.2% of those with type 1 diabetes (T1D) achieved LDL-C levels below 70 mg/dL [44]. Available data from studies using PCSK9 in patients with diabetes are available in Table 3.

Research on omega-3 fatty acids (Eicosapentaenoic Acid—EPA and Docosahexaenoic Acid—DHA) for cardiovascular protection has shown mixed results. The ASCEND and ORIGIN trials, using 840 mg/day omega-3 supplements, failed to reduce major cardiovascular events in people with diabetes, though ASCEND did reduce vascular deaths by 19% [51,52]. The REDUCE-IT trial demonstrated significant benefits using 4 g/day of purified EPA in high-risk patients on statin therapy with elevated triglycerides [53]. Among participants with diabetes, this treatment reduced the primary cardiovascular outcome (CV death, non-fatal myocardial infarction, non-fatal stroke, coronary revascularization, or hospitalization for unstable angina) by 23% and secondary endpoints (CV death, non-fatal myocardial infarction, non-fatal stroke) by 30% [53].

## 3. Current Guidelines

Guidelines addressing LDL-C targets and management in T1D include the 2025 ADA Standards of Medical Care in Diabetes (SOMC) [19], the 2019 European Society of Cardiology/European Atherosclerosis Society (ESC/EAS) Guidelines for Management of Dyslipidaemias [54], and the 2018 ACC/AHA Guidelines on Management of Blood Cholesterol [20]. Pediatric-specific guidelines in T1D include the 2024 International Society of Pediatric and Adolescent Diabetes (ISPAD) Clinical Practice Guidelines [55] and the 2019 AHA Scientific Statement on Cardiovascular Risk Reduction in High-Risk Pediatric Patients [56]. Existing recommendations from these guidelines for the adult population are summarized in Table 4 and for the pediatric population in Table 5.

According to the ADA guidelines, recommendations for adults with T1D are not differentiated from those for T2D; individuals with T1D are included in the same recommendation group as those with T2D who are under 40 years of age. For patients between 20 and 39 years of age, statin therapy is considered if other cardiovascular risk factors are present. For individuals aged 40–75 without a history of ASCVD, statins are recommended in moderate to high doses. For those with high cardiovascular risk or established ASCVD, high-dose statins are recommended with a target of LDL cholesterol below 70 or even below 55 mg/dL. The addition of ezetimibe and PCSK9 inhibitors is considered if the targets are not reached [19]. For the pediatric population with T1D and dyslipidemia, it is recommended that glycemic control and lipid-lowering nutritional therapy be their initial strategy, statins being necessary when LDL-C exceeds 130 mg/dl, with targets below 100 mg/dL.

According to the ESC/EAS dyslipidemia guidelines, patients with T1D are classified as having moderately high to very high cardiovascular risk, depending on diabetes duration, the presence of complications, and coexisting cardiovascular risk factors. LDL-C targets are set according to the degree of risk, and initial therapy typically involves statins, with consideration of ezetimibe and PCSK9 inhibitors if these targets are not achieved. The importance of early intervention with tighter LDL-C targets in patients with a longer duration of type 1 diabetes is emphasized. In the pediatric population, this guideline recommends initiating statins from the age of 10–11 if LDL-C is over 130 mg/dL or if other risk factors are present [54]. It is important to mention that this guideline is stricter in terms of risk classification, LDL targets, and the aggressiveness of lipid-lowering treatment than the 2025 ADA guideline.

The ACC/AHA Guidelines on the Management of Blood Cholesterol have similar recommendations to the 2025 ADA guidelines regarding pediatric patients with T1D or those over 40 years of age. In patients aged 20–39 years, it recommends initiating statins if the duration of T1D is over 20 years, if there are markers of nephropathy (albuminuria, eGFR below 60 mL/min) or other microvascular complications (retinopathy or neuropathy), or if the ankle–brachial index is below 0.9. Recommended LDL-C targets are as strict as those formulated by the ESC/EAS dyslipidemia guidelines [20].

ISPAD advises initiating dyslipidemia screening at age 11 or earlier if additional risk factors are present, with annual follow-up if abnormalities are detected. Statin therapy is recommended for children over 10 years with LDL-C levels >130 mg/dL, aiming for a target below 100 mg/dL [55].

The current guidelines’ recommendations, as mentioned earlier, are inconsistent, with no clear distinction made between recommendations for adult patients with type 1 and type 2 diabetes. This inconsistency is primarily due to the lack of specific evidence for patients with type 1 diabetes and the extrapolation of existing information on the treatment of patients with type 2 diabetes. Recommendations for pediatric patients are consistent but are supported by limited studies and require additional information. These findings again underscore the need for detailed studies on T1D to update the guidelines according to the specific needs of these patients.

## 4. Clinical Implications

A lack of T1D-specific outcome data creates uncertainty for clinicians, potentially contributing to the delayed or underuse of statins in eligible T1D patients [31]. Concerns about polypharmacy in young adults, and insufficient awareness of cardiovascular risk in this population, may contribute as well. Traditional CV risk calculators have not been validated in T1D populations and may underestimate risk, particularly in younger patients. Thus, the development of T1D-specific clinical decision support tools is required to better stratify CV risk and implement treatment.

Integration of continuous glucose monitoring (CGM) in the CV risk assessment represents a possible future direction. Time-in-range metrics and glycemic variability indices may prove superior to HbA1c for predicting cardiovascular outcomes and for guiding lipid management therapy [57,58,59].

Patient-centered considerations include addressing the psychological burden of additional medications in a population already managing complex insulin regimens. Educational interventions emphasizing atherosclerosis and CV risk in T1D, beyond traditional microvascular complications, may be essential for improving treatment acceptance and adherence.

A research priority nowadays is conducting large-scale, adequately powered randomized controlled trials in T1D populations, with cardiovascular endpoints and extended follow-up periods, rather than relying on surrogate markers or extrapolating from mixed-diabetes cohorts. Critical gaps include the development of T1D-specific cardiovascular risk stratification tools and treatment guidelines. Additionally, studies investigating PCSK9 inhibitors specifically in T1D populations, integrating CGM metrics with CV risk assessment, and addressing guidelines and limitations to optimal statin use in eligible patients with T1D are essential to improve evidence-based care and cardiovascular outcomes in this high-risk population.

## 5. Limitations

This study represents a narrative review, as several key methodological criteria for systematic reviews were not implemented. While two reviewers independently performed the literature search, the absence of formal inter-rater reliability assessment and standardized quality appraisal tools limits the methodological rigor compared to systematic reviews. These limitations were considered acceptable given our objective to synthesize the existing evidence and clarify the current knowledge in a field that still lacks sufficient scientific data.

Current evidence for dyslipidemia management in T1D faces several constraints that limit clinical applicability and guideline recommendations. There is a limited number of studies that include the T1D population. The majority of lipid-lowering outcome trials have focused on T2D or mixed populations, with T1D patients comprising less than 5% of participants in landmark studies, such as IMPROVE-IT and FOURIER [36,46]. Sample size constraints further compromise statistical power, particularly for subgroup analysis.

Meanwhile, conventional LDL-C targets may inadequately capture cardiovascular risk in patients with T1D since the pathogenic pathways of atherosclerosis are different in T1D compared with T2D.

Pediatric evidence presents even greater limitations, with most studies relying on surrogate endpoints rather than hard cardiovascular outcomes. Meanwhile, the relatively short follow-up periods (typically 2–4 years) in existing studies are insufficient to capture the long-term cardiovascular benefits.

## 6. Methods

We conducted a literature review to assess the current evidence and recommendations on the treatment of dyslipidemia in patients with T1D. There were two reviewers performing the search independently. A comprehensive search was performed using electronic medical databases (PubMed, Embase, and the Cochrane Library), covering studies published from January 2000 to June 2025. Search terms included combinations of the following: “type 1 diabetes”, “dyslipidemia,” “lipid-lowering therapy”, “statins”, “ezetimibe”, “PCSK9 inhibitors”, “cardiovascular disease”, and “clinical guidelines”. Additional sources were identified by manual screening of reference lists from key reviews, guidelines, and primary studies.


Studies were included if they met the following criteria:



Investigating lipid management in patients with T1D, T2D, or both;Clinical trials (randomized or non-randomized), cohort studies, meta-analyses, or systematic reviews;Containing data on lipid parameters, cardiovascular outcomes, or treatment efficacy;Clinical practice guidelines from major international organizations, such as the American Diabetes Association (ADA), the European Society of Cardiology (ESC)/European Atherosclerosis Society (EAS), the American College of Cardiology (ACC), the American Heart Association (AHA), and the International Society for Pediatric and Adolescent Diabetes (ISPAD).


Articles not available in English, abstracts without full texts, and single-case reports were excluded. Given the heterogeneity of study designs and outcome measures, the results were synthesized narratively. Emphasis was placed on identifying evidence specific to T1D and variations among current international guidelines (Figure 1).

## 7. Conclusions

Adults with T1D have an increased cardiovascular risk. Additionally, there is evidence supporting that proatherogenic changes appear in these patients even from childhood to adolescence. Studies suggest that LDL-C is the most important modifiable risk factor for ASCVD in T1D, even in patients with controlled glycemia. Statin therapy remains the first-line treatment, as suggested by existing studies and recommended by current guidelines. However, the results specific to T1D populations are inconclusive, especially in pediatric groups, where the benefits on surrogate markers, such as arterial stiffness or intima–media thickness, are not clear. Other therapies that have been proven efficient in reducing LDL-C, with potential cardiovascular benefits, are ezetimibe and PCSK9 inhibitors. Notably, the latest data suggest an enhanced efficacy of ezetimibe in T1D, potentially due to its differences in cholesterol metabolism. Current guidelines offer variable recommendations for adults with T1D. Pediatric guidelines are accurate but need more evidence from RCTs to support the recommendations. Such data is essential to support precise, evidence-based recommendations and to cover the existing knowledge gap between type 1 and type 2 diabetes management.

## Figures and Tables

**Figure 1 ijms-26-08558-f001:**
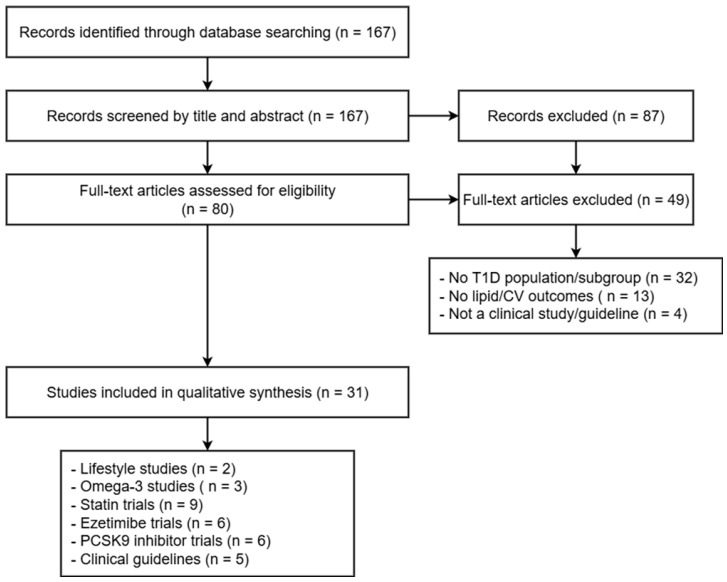
Flow diagram of study selection.

**Table 1 ijms-26-08558-t001:** Summary of statin therapy studies in diabetes.

SN	Author and Year	Title	Primary and Secondary Objectives	Population/Participants	Sample Size	Intervention/Exposure	Outcome Measures	Findings and Conclusions
1	Colhoun, H.M. et al., 2004 [24]	Primary prevention of cardiovascular disease with atorvastatin in T2D in the Collaborative Atorvastatin Diabetes Study (CARDS)	Primary: to assess the effect of atorvastatin on cardiovascular events in T2D. Secondary: to evaluate safety and specific cardiovascular outcomes.	Type 2 diabetes with no prior cardiovascular disease	2838	Atorvastatin (10 mg/day) vs. placebo	Major cardiovascular events (acute coronary events, stroke, revascularization)	Atorvastatin reduced major cardiovascular events by 37% (HR 0.63, 95% CI 0.48–0.83), with a favorable safety profile.
2	Collins, R. et al., 2002 [25]	MRC/BHF Heart Protection Study of cholesterol lowering with simvastatin in 20,536 high-risk individuals	Primary: to assess the effect of simvastatin on major vascular events in high-risk individuals. Secondary: to assess effects in subgroups, including patients with diabetes.	High-risk individuals, including T1D and T2D	20,536	Simvastatin (40 mg/day) vs. placebo	Major vascular events (coronary events, stroke, revascularization)	Simvastatin reduced major vascular events by 22% in patients with diabetes (RR 0.78, 95% CI 0.67–0.91) vs. placebo.
3	Downs, J.R. et al., 2008 [26]	Efficacy of cholesterol-lowering therapy in 18,686 people with diabetes in 14 randomized trials of statins: a meta-analysis	Primary: to assess the effect of statins on CV outcomes in patients with diabetes. Secondary: to evaluate effects on specific vascular events and mortality.	Diabetes patients (type 1 and 2) from 14 randomized trials	18,686	Statin therapy vs. control	Major vascular events, coronary events, stroke, mortality	Statins reduced major vascular events by 21% per 1 mmol/L of LDL-C reduction (RR 0.79, 95% CI 0.72–0.86), with no significant difference between type 1 and type 2 diabetes.
4	Nathan, D.M. et al., 2005 [27]	Intensive diabetes treatment and cardiovascular disease in patients with T1D	Primary: to assess the effect of intensive diabetes treatment on CV disease in T1D. Secondary: to assess long-term complications.	Type 1 diabetes patients	1441	Intensive vs. conventional diabetes therapy	Cardiovascular events (myocardi-al infarction, stroke, cardiovascular death)	Intensive treatment reduced cardiovascular events by 42% (HR 0.58, 95% CI 0.39–0.86) in type 1 diabetes patients.
5	Martin, S. et al., 2011 [33]	Residual beta cell function in newly diagnosed type 1 diabetes after treatment with atorvastatin: the Randomized DIATOR Trial	Primary: to assess the effect of atorvastatin on residual beta-cell function in newly diagnosed T1D. Secondary: to evaluate inflammatory markers.	Newly diagnosed patients with T1D	89	Atorvastatin (80 mg/day) vs. placebo	C-peptide levels, inflammatory markers (CRP, IL-6)	Atorvastatin did not significantly preserve beta-cell function (C-peptide levels) or reduce inflammatory markers
6.	de Zeeuw, D. et al., 2015 [34]	Renal effects of atorvastatin and rosuvastatin in patients with diabetes who have progressive renal disease (PLANET I)	Primary: to compare the renal effects of atorvastatin and rosuvastatin in diabetes patients with proteinuria. Secondary: to assess lipid-lowering efficacy.	Diabetes patients with progressive renal disease	353	Atorvastatin (80 mg/day) vs. rosuvastatin (10 or 40 mg/day)	Urinary protein/creatinine ratio, glomerular filtration rate, LDL-C	Atorvastatin reduced proteinuria more effectively than rosuvastatin, with similar LDL-C reduction but better renal outcomes.
7	De Vries, F.M. et al., 2014 [30]	Efficacy of Standard and Intensive Statin Treatment for the Secondary Prevention of Cardiovascular and Cerebrovascular Events in Diabetes Patients: A Meta-Analysis	Primary: to compare standard vs. intensive statin therapy for secondary cardiovascular prevention in diabetes. Secondary: to assess cerebrovascular events.	Diabetes patients with prior cardiovascular disease	12,563	Standard vs. intensive statin therapy	Cardiovascular and cerebrovascular events	Intensive statin therapy reduced cardiovascular events by 9% (RR 0.90, 95% CI 0.85–0.96) compared to standard therapy in diabetes patients.
8	Yoo, J. et al., 2023 [31]	Impact of statin treatment on cardiovascular risk in patients with type 1 diabetes: a population-based cohort study	Primary: to evaluate the effect of statins on CV risk in T1D. Secondary: to assess specific CV outcomes.	Type 1 diabetes patients	11,009	Statin use vs. non-use	Major adverse cardiovascular events (MACEs)	Statin use was associated with a 40% reduction in MACEs (HR 0.60, 95% CI 0.45–0.80)
9.	Marcovecchio, M.L. et al., 2017 [29]	ACE Inhibitors and Statins in Adolescents with Type 1 Diabetes Adolescent type 1 Diabetes cardio-renal Intervention Trial (AdDIT)	Primary: to evaluate the change in albumin excretion in adolescents with T1D. Secondary: to assess the development of microalbuminuria, progression of retinopathy, changes in eGFR, lipid levels, and CV risk.	Adolescents with T1D	443	Statins and/or ACE inhibitors vs. placebo	Carotid intima–media thickness, albuminuria, lipid levels	Statin use resulted in significant reductions in TC, LDL-C, and non-HDL-C, in triglyceride levels, and in the ratio of apo B to apo A1.
10.	Haller, M.J. et al., 2009 [32]	Pediatric Atorvastatin in Diabetes Trial (PADIT): a pilot study to determine the effect of atorvastatin on arterial stiffness and endothelial function in children with type 1 diabetes	Primary: to assess atorvastatin’s effect on arterial stiffness and endothelial function in children with T1D. Secondary: to evaluate safety and lipid profile changes.	Children and adolescents (10–21 years old) with T1D	51	Atorvastatin (20 mg/day) vs. placebo	Pulse wave velocity (arterial stiffness), FMD (endothelial function), lipid profiles	Atorvastatin improved endothelial function (*p* = 0.04) but did not significantly affect arterial stiffness; it was safe and reduced LDL-C levels by 25%.
11	Hero, C. et al., 2016 [28]	Association between use of lipid-lowering therapy and cardiovascular diseases and death in individuals with type 1 diabetes	Primary: to investigate the association between lipid-lowering therapy and cardiovascular events/death in T1D. Secondary: to assess specific cardiovascular outcomes and mortality rates.	Individuals with type 1 diabetes from the Swedish National Diabetes Register	24,230	Lipid-lowering therapy vs. no lipid-lowering therapy	Cardiovascular events (myocardial infarction, stroke, coronary heart disease), all-cause mortality	Lipid-lowering therapy was associated with a 22–44% reduction in cardiovascular events and death (HR 0.78 for cardiovascular disease, 95% CI 0.68–0.89; HR 0.56 for mortality, 95% CI 0.48–0.66).

Abbreviations: T2D: type 2 diabetes, T1D: type 1 diabetes, CV: cardiovascular, HR: hazard ratio, CI: confidence interval, RR: relative risk, LDL-C: low-density lipoprotein cholesterol, CRP: C-Reactive Protein, IL-6: Interleukin-6, eGFR: estimated Glomerular Filtration Rate, MACE: major adverse cardiovascular event, TC: total cholesterol, non-HDL: non-High-Density Lipoprotein, apo B: apolipoprotein B, apo A1: apolipoprotein A1, FMD: flow-mediated dilation.

**Table 2 ijms-26-08558-t002:** Summary of ezetimibe studies in diabetes.

SN	Author and Year	Title	Primary and Secondary Objectives	Population/Participants	Sample Size	Intervention/Exposure	Outcome Measures	Findings and Conclusions
1	Giugliano, R.P. et al., 2018 [36]	Benefit of Adding Ezetimibe to Statin Therapy on Cardiovascular Outcomes and Safety in Patients with Versus Without Diabetes Mellitus	Primary: to assess if ezetimibe plus simvastatin reduces cardiovascular events compared to simvastatin alone. Secondary: to evaluate safety and efficacy in diabetes vs. non-diabetes patients.	Patients with acute coronary syndrome, with or without diabetes mellitus	18,144	Ezetimibe plus simvastatin vs. simvastatin alone	Primary: Composite of cardiovascular death, myocardial infarction, stroke, or revascularization. Secondary: Safety endpoints (adverse events).	Ezetimibe plus simvastatin significantly reduced cardiovascular events in patients with diabetes (HR 0.86, 95% CI 0.78–0.94) compared to simvastatin alone, with consistent safety profiles.
2	Lee, Y.J. et al., 2023 [37]	Moderate-intensity statin with ezetimibe vs. high-intensity statin in patients with diabetes and atherosclerotic cardiovascular disease in the RACING trial	Primary: to compare cardiovascular outcomes of moderate-intensity statin plus ezetimibe vs. high-intensity statin. Secondary: to assess safety and LDL-C reduction.	Patients with diabetes and atherosclerotic cardiovascular disease	3780	Moderate-intensity statin plus ezetimibe vs. high-intensity statin	Primary: Composite of cardiovascular death, major cardiovascular events, or stroke. Secondary: LDL-C levels, adverse events.	Moderate-intensity statin plus ezetimibe was non-inferior to high-intensity statin in reducing cardiovascular events (HR 0.94, 95% CI 0.82–1.09), with better LDL-C reduction and fewer adverse events.
3	Kim, B.K. et al., 2022 [38]	Long-term efficacy and safety of moderate-intensity statin with ezetimibe versus high-intensity statin monotherapy in patients with atherosclerotic cardiovascular disease (RACING)	Primary: to assess the non-inferiority of moderate-intensity statin plus ezetimibe vs. high-intensity statin for cardiovascular outcomes. Secondary: to assess long-term safety and tolerability.	Patients with atherosclerotic cardiovascular disease	3780	Moderate-intensity statin plus ezetimibe vs. high-intensity statin monotherapy	Primary: Composite of cardiovascular death, major cardiovascular events, or non-fatal stroke. Secondary: Adverse events, LDL-C levels.	Moderate-intensity statin plus ezetimibe was non-inferior (HR 0.92, 95% CI 0.80–1.05), with lower rates of intolerance-related discontinuations.
4	Yunoki, K. et al., 2011 [39]	Ezetimibe improves postprandial hyperlipemia and its induced endothelial dysfunction	Primary: to assess the effect of ezetimibe on postprandial hyperlipidemia and endothelial function. Secondary: to assess changes in lipid profiles.	Patients with dyslipidemia	20	Ezetimibe (10 mg/day) vs. placebo	Postprandial triglyceride levels, flow-mediated dilatation (FMD), lipid profiles	Ezetimibe significantly reduced postprandial triglyceride levels and improved FMD, indicating better endothelial function.
5	Ciriacks, K. et al., 2015 [40]	Effects of simvastatin and ezetimibe in lowering low-density lipoprotein cholesterol in subjects with type 1 and type 2 diabetes mellitus	Primary: to compare LDL-C reduction with simvastatin vs. simvastatin plus ezetimibe. Secondary: to assess effects on other lipid parameters.	Patients with T1D and T2D	40	Simvastatin (40 mg/day) vs. simvastatin plus ezetimibe (10 mg/day)	LDL-C levels, other lipid parameters (HDL-C, triglycerides)	Simvastatin plus ezetimibe resulted in greater LDL-C reduction (*p* < 0.05) compared to simvastatin alone in both patients with T1D and T2D.
6	Semova, I. et al., 2019 [41]	Type 1 diabetes is associated with an increase in cholesterol absorption markers but a decrease in cholesterol synthesis markers in a young adult population	Primary: to assess cholesterol metabolism in T1D. Secondary: to compare absorption and synthesis markers between T1D and controls.	Young adults with T1D and healthy controls	200	Observational (no intervention)	Cholesterol absorption (campesterol, sitosterol) and synthesis (lathosterol) markers	Type 1 diabetes patients had higher cholesterol absorption markers and lower synthesis markers compared to controls, suggesting altered cholesterol metabolism.

Abbreviations: HR: hazard ratio, CI: confidence interval, LDL-C: low-density lipoprotein cholesterol, FMD: flow-mediated dilatation, HDL-C: High-Density Lipoprotein Cholesterol, T1D: type 1 diabetes, T2D: type 2 diabetes.

**Table 3 ijms-26-08558-t003:** Summary of PCSK9-related therapy studies in diabetes.

SN	Author and Year	Title	Primary and Secondary Objectives	Population/Participants	Sample Size	Intervention/Exposure	Outcome Measures	Findings and Conclusions
1	Kang, Y.M. et al., 2025 [47]	Cardiovascular Outcomes and Efficacy of the PCSK9 Inhibitor Evolocumab in Individuals With Type 1 Diabetes: Insights From the FOURIER Trial	Primary: to assess the effect of evolocumab on cardiovascular outcomes in T1D. Secondary: to assess safety and LDL-C reduction.	Patients with T1D and high CV risk	27,564 participants, of which 10,834 had T2D and 197 had T1D	Evolocumab vs. placebo	MACEs (CV death, myocardial infarction, stroke, hospitalization for unstable angina, or coronary revascularization), LDL-C levels, safety endpoints	Evolocumab reduced MACEs in patients with T1D by 34% (HR = 0.66, CI = 0.32–1.38) with significant LDL-C reduction, and a similar safety profile to the overall cohort.
2	Cariou, B. et al., 2017 [50]	Efficacy and safety of alirocumab in insulin-treated patients with type 1 or type 2 diabetes and high cardiovascular risk: Rationale and design of the ODYSSEY DM–INSULIN trial	Primary: to assess the efficacy of alirocumab in reducing LDL-C in insulin-treated diabetes patients. Secondary: to evaluate safety and other lipid parameters.	Insulin-treated patients with type 1 or type 2 diabetes with high CV risk	517 (planned)	Alirocumab vs. placebo	LDL-C reduction, adverse events, other lipid parameters	Ongoing trial.
3	Levenson, A.E. et al., 2017 [42]	PCSK9 Is Increased in Youth With Type 1 Diabetes	Primary: to compare PCSK9 levels in youth with type 1 diabetes vs. controls. Secondary: to assess associations with glycemic control and lipids.	Youth with T1D and healthy controls	70	Observational (no intervention)	PCSK9 levels, HbA1c, lipid profiles	PCSK9 levels were significantly higher in youth with T1D (*p* < 0.01) and correlated with HbA1c, suggesting a link to glycemic control.
4	Bojanin, D. et al., 2019 [43]	Association between PCSK9 and lipoprotein subclasses in children with T1D: effects of glycemic control	Primary: to investigate PCSK9′s association with lipoprotein subclasses in children with T1D. Secondary: to evaluate the effects of glycemic control.	Children with T1D	60	Observational (no intervention)	PCSK9 levels, lipoprotein subclasses, HbA1c	Higher PCSK9 levels were associated with adverse lipoprotein profiles in T1D, with better glycemic control linked to lower PCSK9 levels.
5	O’Donoghue, M.L. et al., 2019 [46]	Lipoprotein(a), PCSK9 inhibition, and cardiovascular risk insights from the FOURIER trial	Primary: to assess the effect of evolocumab on lipoprotein (a) and CV risk. Secondary: to evaluate outcomes in subgroups, including diabetes.	Patients with ASCVD (subgroup with diabetes)	27,564	Evolocumab vs. placebo	Lipoprotein (a) levels, CV events	Evolocumab reduced lipoprotein (a) by 26.9% and cardiovascular events, with consistent benefits in patients with diabetes.
6	Bittner, V.A. et al., 2020 [48]	Effect of Alirocumab on Lipoprotein(a) and Cardiovascular Risk After Acute Coronary Syndrome	Primary: to evaluate the effect of alirocumab on lipoprotein (a) and CV risk post-acute coronary syndrome Secondary: to assess outcomes in diabetes subgroups.	Patients with post-acute coronary syndrome, including those with diabetes	18,924	Alirocumab vs. placebo	Lipoprotein (a) levels, major adverse CV events	Alirocumab reduced lipoprotein (a) by 23% and CV events by 15% (HR 0.85, 95% CI 0.78–0.93), with benefits observed in patients with diabetes.

Abbreviations: MACE: major adverse cardiovascular event, HR: hazard ratio, CI: confidence interval, LDL-C: low-density lipoprotein cholesterol, T1D: type 1 diabetes, T2D: type 2 diabetes, CV: cardiovascular, HbA1c: Hemoglobin A1c, PCSK9: Proprotein Convertase Subtilisin/Kexin Type 9, ASCVD: atherosclerotic cardiovascular disease.

**Table 4 ijms-26-08558-t004:** Current guidelines and recommendations for LDL-C lowering in type 1 diabetes in adults.

Adult Guidelines						
Guideline	Organization	Age Group	Risk Classification	Statin Indication	LDL-C Target	Additional Therapy
2025 ADA SOMC [19]	American Diabetes Association	20–39 years	Same as T2D under 40	If other CV risk factors are associated	Not specified	Ezetimibe and PCSK9 inhibitors if targets not reached
		40–75 years without ASCVD	Same as T2D	Moderate-high dose statins recommended	Not specified	
		High CV risk or established ASCVD	Same as T2D	High-dose statins recommended	<70 mg/dL or <55 mg/dL	
2019 ESC/EAS [54]	European Society of Cardiology/European Atherosclerosis Society	All adults	Moderately high or very high CV risk (depending on diabetes duration, complications, other CV risk factors)	Statins as initial therapy	Risk-stratified targets	Ezetimibe and PCSK9 inhibitors if targets not reached
2018 ACC/AHA [20]	American College of Cardiology/American Heart Association	20–39 years	Similar to the 2025 ADA	If T1D duration > 20 years, nephropathy markers, microvascular complications, or ABI < 0.9	Same as ESC/EAS (strict targets)	Similar to other guidelines
		>40 years	Similar to the 2025 ADA	Similar to the 2025 ADA		

**Table 5 ijms-26-08558-t005:** Current guidelines and recommendations for LDL-C lowering in type 1 diabetes in children.

Pediatric Guidelines						
Guideline	Organization	Age Group	Screening	Statin Indication	LDL-C Target	Initial Strategy
2025 ADA SOMC [19]	American Diabetes Association	Pediatric	Not specified	When LDL-C > 130 mg/dL	<100 mg/dL	Glycemic control and lipid-lowering nutritional therapy
2019 ESC/EAS [54]	European Society of Cardiology/European Atherosclerosis Society	10–11+ years	Not specified	If LDL-C > 130 mg/dL or other risk factors are present	Not specified	Statins from age 10 to 11
2018 ACC/AHA [20]	American College of Cardiology/American Heart Association	Pediatric	Similar to the 2025 ADA	Similar to the 2025 ADA	Similar to the 2025 ADA	Similar to the 2025 ADA
2024 ISPAD [55]	International Society of Pediatric and Adolescent Diabetes	11+ years (earlier if risk factors are associated)	Annual screening if values have changed	Children > 10 years with LDL-C > 130 mg/dL	<100 mg/dL	Screening starting at age 11
2019 AHA Scientific Statement [56]	American Heart Association	High-risk pediatric patients	Not specified	CV risk reduction focus	Not specified	Risk reduction strategies

Abbreviations: SOMC: Standards of Medical Care in Diabetes; CV: cardiovascular; ASCVD: atherosclerotic cardiovascular disease; ABI: ankle–Brachial index; eGFR: estimated Glomerular Filtration Rate, ADA: American Diabetes Association, ESC: European Society of Cardiology, EAS: European Atherosclerosis Society, ACC: American College of Cardiology, AHA: American Heart Association, ISPAD: International Society of Pediatric and Adolescent Diabetes, LDL-C: low-density lipoprotein cholesterol, T1D: type 1 diabetes, T2D: type 2 diabetes.

## Data Availability

The original contributions presented in this study are included in the article. Further inquiries can be directed to the corresponding author.

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
