# Peer review of "Treatment of Dyslipidemia in Patients with Type 1 Diabetes Mellitus: A Review of Current Evidence and Knowledge Gaps"

_ijms, 2025, doi:10.3390/ijms26178558_

Round 1

Reviewer 1 Report

Comments and Suggestions for Authors

Dear authors,

It was a pleasure to review your article entitled “Treatment of Dyslipidemia in Patients with Type 1 Diabetes Mellitus: A Review of Current Evidence and Knowledge Gaps.” The manuscript is a systematic review that synthesizes current evidence on how lipid behavior affects cardiovascular risk in people with type 1 diabetes and updates the main therapeutic options.

Introduction —Please add a paragraph that describes lipid metabolism in type 1 diabetes in greater detail, contrasting it with insulin-resistant states/type 2 diabetes and explaining why cardiovascular risk emerges earlier in this population.

Methods — The manuscript does not specify: the number of reviewers and screening stages; how selection bias was addressed; which risk-of-bias tool was used (e.g., RoB 2, ROBINS-I, QUADAS-2); the MeSH terms applied to each database search; the inter-rater agreement statistic (e.g., kappa).

Discussion — Consider adding a section focused on methodological limitations and emerging evidence, highlighting the clinical implications of your findings.

Article type — Although submitted as a “review article,” the methods match those of a “systematic review”. If you prefer to present it as a narrative review, adjust the wording accordingly.

Author Response

Thank you for reviewing our article and for your pertinent comments. 

R: Introduction —Please add a paragraph that describes lipid metabolism in type 1 diabetes in greater detail, contrasting it with insulin-resistant states/type 2 diabetes and explaining why cardiovascular risk emerges earlier in this population.

A: Based on your comment, we have added this information in the introduction, as follows:

Pathophysiology of lipid disorders in T1D fundamentally differs from T2D and insulin-resistance conditions. Thus in T1D absolute insulin deficiency impairs lipoprotein lipase activity, leading to decreased clearance of triglyceride-rich lipoproteins (chylomicrons and VLDL), rather than increased hepatic VLDL production, which is characteristic of insulin resistance [4] . In T2D, insulin resistance stimulates hepatic lipogenesis through persistent SREBP-1c activation, and increased free fatty acid from adipose tissue lipolysis, resulting in the classic triad of hypertriglyceridemia, low HDL-cholesterol, and small dense LDL particles [5,6,7].

CV risk emerges earlier in T1D through multiple insulin deficiency-independent pathways, including hyperglycemia-induced advanced glycation end products (AGEs) formation, enhanced myelopoiesis with calprotectin release stimulating RAGE receptors, with modified glycated and oxidized LDL particles and accelerated atherosclerosis [8]. Chronic hyperglycemia induces AGEs formation that leads to NADPH oxidase activation, mitochondrial reactive oxygen species generation, and subsequent endothelial dysfunction through decreased nitric oxide bioavailability [9]. This process is amplified by protein kinase C activation and increased polyol pathway flux, generating oxidative stress and accelerating atherogenesis [10]. The autoimmune process involved in T1D contributes to an additional inflammatory burden, with calprotectin release from activated neutrophils promoting myelopoiesis through bone marrow RAGE receptor [11,12].

Peripheral hyperinsulinemia from subcutaneous insulin administration can lead to qualitative HDL abnormalities and altered lipoprotein composition that may compromise some cardioprotective functions [2]. These molecular changes explain why even well-controlled patients with T1D exhibit qualitative lipoprotein abnormalities that persist independently of glycemic status [13,14].

R: Methods — The manuscript does not specify: the number of reviewers and screening stages; how selection bias was addressed; which risk-of-bias tool was used (e.g., RoB 2, ROBINS-I, QUADAS-2); the MeSH terms applied to each database search; the inter-rater agreement statistic (e.g., kappa).

A: Thank you for this comment. Now, we mentioned that two reviewers (V.E and A.D.) have separately assessed the literature. Since this is a narrative review, we did not use any other tools which are specific to a systematic review. This is now clearly mentioned in the limitations section.

R: Discussion — Consider adding a section focused on methodological limitations and emerging evidence, highlighting the clinical implications of your findings.

A: Thank you for this comment. We have added this information as follows:

  1. Limitations

This study represents a narrative review, as several key methodological criteria for systematic reviews were not implemented. While two reviewers independently performed the literature search, the absence of formal inter-rater reliability assessment and standardized quality appraisal tools limits the methodological rigor compared to systematic reviews. These limitations were considered acceptable given our objective to synthesize the existing evidence and clarify the current knowledge in a field that still lacks sufficient scientific data.

Current evidence for dyslipidemia management in T1D faces several constraints that limit clinical applicability and guideline recommendations. There is a limited number of studies that include T1D population. Majority of lipid-lowering outcome trials have focused on T2D or mixed populations, with T1D patients comprising less than 5% of participants in landmark studies such as IMPROVE-IT and FOURIER [36,46]. Sample size constraints further compromise statistical power, markedly for subgroup analysis.

Conventional LDL-C targets may inadequately capture cardiovascular risk in T1D patients, since the pathogenic pathways of atherosclerosis are different in T1D compared with T2D.

Pediatric evidence presents even greater limitations, with most studies relying on surrogate endpoints rather than hard cardiovascular outcomes. Meanwhile, the relatively short follow-up periods (typically 2-4 years) in existing studies are insufficient to capture the long-term cardiovascular benefits.

  1. Clinical Implications

Lack of T1D-specific outcome data creates uncertainty for clinicians, potentially contributing to the delayed or underuse of statins in eligible T1D patients [31]. Concerns about polypharmacy in young adults, and insufficient awareness of cardiovascular risk in this population may contribute as well. Traditional CV risk calculators have not been validated in T1D populations and may underestimate risk, particularly in younger patients.  Thus, development of T1D-specific clinical decision support tools is required to better stratify CV risk and implement treatment.

Integration of continuous glucose monitoring (CGM) in the CV risk assessment represents a possible future direction. Time-in-range metrics and glycemic variability indices may prove superior to HbA1c for predicting cardiovascular outcomes and for guiding lipid management therapy [56, 57, 58].

Patient-centered considerations include addressing the psychological burden of additional medications in a population already managing complex insulin regimens. Educational interventions emphasizing atherosclerosis and CV risk in T1D, beyond traditional microvascular complications, may be essential for improving treatment ac-acceptance and adherence.

A research priority nowadays is conducting large-scale, adequately powered randomized controlled trials in T1D populations, with cardiovascular endpoints and ex-tended follow-up periods, rather than relying on surrogate markers or extrapolating from mixed-diabetes cohorts. Critical gaps include development of T1D-specific cardiovascular risk stratification tools and treatment guidelines. Additionally, studies investigating PCSK9 inhibitors specifically in T1D populations, research integrating CGM metrics with CV risk assessment, and address guidelines limitations to optimal statin use in eligible patients with T1D are essential to improve evidence-based care and cardio-vascular outcomes in this high-risk population.

R: Article type — Although submitted as a “review article,” the methods match those of a “systematic review”. If you prefer to present it as a narrative review, adjust the wording accordingly.

A: Based on your comment we clearly specified that this is a narrative review.

While two reviewers (V.E. and A.D.) independently performed the literature search, the absence of formal inter-rater reliability assessment and standardized quality appraisal tools limits the methodological rigor compared to systematic reviews. This is now included in the limitations section.

These limitations were considered acceptable given our objective to synthesize the existing evidence, and clarify the current knowledge in a field that still lacks sufficient scientific data.

Reviewer 2 Report

Comments and Suggestions for Authors

Overall, the manuscript is well-written, and the authors have made a commendable effort to provide a comprehensive summary on the treatment of dyslipidemia in patients with Type 1 Diabetes Mellitus. However, a few points should be addressed before publication:

  1. The abstract should clearly specify the range of years of the studies reviewed.

  2. The reference formatting should follow the MDPI guidelines consistently.

  3. The plagiarism similarity index should be reduced to below 15% for this review.

  4. In the section discussing mechanisms, I did not find sufficient observational evidence; this should be incorporated into the manuscript.

  5. The limitations related to treatment approaches and study design should be explicitly stated.

Overall, the manuscript successfully addresses the main topic of the paper, but these revisions are necessary to meet publication standards.

Author Response

Thank you for reviewing our article and for your pertinent comments. 

R: The abstract should clearly specify the range of years of the studies reviewed.

A: We have revised the abstract and mentioned this information

R: The reference formatting should follow the MDPI guidelines consistently.

A: We have corrected the References accordingly

R: The plagiarism similarity index should be reduced to below 15% for this review.

A: We have checked the similarities and mentioned the sources in the text

R: In the section discussing mechanisms, I did not find sufficient observational evidence; this should be incorporated into the manuscript.

A: We have added this information in the introduction. You can find the paragraph inserted below:

Pathophysiology of lipid disorders in T1D fundamentally differs from T2D and in-sulin-resistance conditions. Thus, in T1D absolute insulin deficiency impairs lipoprotein lipase activity, leading to decreased clearance of triglyceride-rich lipoproteins (chylo-microns and VLDL), rather than increased hepatic VLDL production, which is charac-teristic of insulin resistance [4] . In T2D, insulin resistance stimulates hepatic lipogenesis through persistent SREBP-1c activation, and increased free fatty acid from adipose tissue lipolysis, resulting in the classic triad of hypertriglyceridemia, low HDL-cholesterol, and small dense LDL particles [5,6,7].

CV risk emerges earlier in T1D through multiple insulin deficiency-independent pathways, including hyperglycemia-induced advanced glycation end products (AGEs) formation, enhanced myelopoiesis with calprotectin release stimulating RAGE recep-tors, with modified glycated and oxidized LDL particles and accelerated atherosclerosis [8]. Chronic hyperglycemia induces AGEs formation that leads to NADPH oxidase ac-tivation, mitochondrial reactive oxygen species generation, and subsequent endothelial dysfunction through decreased nitric oxide bioavailability [9]. This process is amplified by protein kinase C (PKC) activation and stimulated polyol pathway, generating oxida-tive stress and accelerating atherogenesis [10]. The autoimmune process involved in T1D contributes to an additional inflammatory burden, with calprotectin release from acti-vated neutrophils promoting myelopoiesis through bone marrow RAGE receptor [11,12].

Peripheral hyperinsulinemia from subcutaneous insulin administration can lead to qualitative HDL abnormalities and altered lipoprotein composition that may compro-mise some cardioprotective functions [2]. These molecular changes explain why even well-controlled patients with T1D exhibit qualitative lipoprotein abnormalities that persist independently of glycemic status [13,14].

R: The limitations related to treatment approaches and study design should be explicitly stated.

A: We have added information on study design limitations. You can find the paragraph inserted below:

  1. Limitations

This study represents a narrative review, as several key methodological criteria for systematic reviews were not implemented. While two reviewers independently performed the literature search, the absence of formal inter-rater reliability assessment and standardized quality appraisal tools limits the methodological rigor compared to systematic reviews. These limitations were considered acceptable given our objective to synthesize the existing evidence and clarify the current knowledge in a field that still lacks sufficient scientific data.

Current evidence for dyslipidemia management in T1D faces several constraints that limit clinical applicability and guideline recommendations. There is a limited number of studies that include T1D population. Majority of lipid-lowering outcome trials have focused on T2D or mixed populations, with T1D patients comprising less than 5% of participants in landmark studies such as IMPROVE-IT and FOURIER [36,46]. Sample size constraints further compromise statistical power, markedly for subgroup analysis.

Meanwhile conventional LDL-C targets may inadequately capture cardiovascular risk in patients with T1D, since the pathogenic pathways of atherosclerosis are different in T1D compared with T2D.

Pediatric evidence presents even greater limitations, with most studies relying on surrogate endpoints rather than hard cardiovascular outcomes. Meanwhile, the relatively short follow-up periods (typically 2-4 years) in existing studies are insufficient to capture the long-term cardiovascular benefits.

Round 2

Reviewer 1 Report

Comments and Suggestions for Authors

Dear Authors,

Thank you for submitting the revised version of your manuscript  "Treatment of Dyslipidemia in Patients with Type 1 Diabetes Mellitus: A Review of Current Evidence and Knowledge Gaps". After reviewing your revisions and point‑by‑point responses, I confirm that the requests have been satisfactorily addressed.
Accordingly, I am pleased to approve the manuscript to proceed to publication, pending only minor production suggestion: Harmonize abbreviations and define them at first mention. 

Author Response

Thank you for reviewing our article and for your comment.

R: Harmonize abbreviations and define them at first mention.

A: In order to follow your recommendation, we have rechecked our paper and we have defined all abbreviations at first mention in the text, and they were all harmonized accordingly.